# Role of Glycolysis/Gluconeogenesis and HIF-1 Signaling Pathways in Rats with Dental Fluorosis Integrated Proteomics and Metabolomics Analysis

**DOI:** 10.3390/ijms23158266

**Published:** 2022-07-27

**Authors:** Yue Ba, Shuo Yang, Shuiyuan Yu, Xiangbo Hou, Yuhui Du, Minghui Gao, Juan Zuo, Lei Sun, Xiaoli Fu, Zhiyuan Li, Hui Huang, Guoyu Zhou, Fangfang Yu

**Affiliations:** Department of Environmental Health, School of Public Health, Zhengzhou University, 100 Kexue Avenue, Zhengzhou 450001, China; byyue@zzu.edu.cn (Y.B.); yang___shuo999@163.com (S.Y.); ysy599926@163.com (S.Y.); houxb2022@163.com (X.H.); honeygirl1112@163.com (Y.D.); mavis_gmh@163.com (M.G.); zuojuan719920@163.com (J.Z.); sl349496017@163.com (L.S.); xlfu66@126.com (X.F.); zhiyuanli@zzu.edu.cn (Z.L.); huihuang@zzu.edu.cn (H.H.); zhouguoyu@zzu.edu.cn (G.Z.)

**Keywords:** glycolysis/gluconeogenesis pathway, HIF-1 pathway, fluoride, dental fluorosis

## Abstract

Fluoride is widely distributed, and excessive intake will lead to dental fluorosis. In this study, six offspring rats administrated 100 mg/L sodium fluoride were defined as the dental fluorosis group, and eight offspring rats who received pure water were defined as the control group. Differentially expressed proteins and metabolites extracted from peripheral blood were identified using the liquid chromatography tandem mass spectrometry and gas chromatography mass spectrometry, with the judgment criteria of fold change >1.2 or <0.83 and *p* < 0.05. A coexpression enrichment analysis using OmicsBean was conducted on the identified proteins and metabolites, and a false discovery rate (FDR) < 0.05 was considered significant. Human Protein Atlas was used to determine the subcellular distribution of hub proteins. The Gene Cards was used to verify results. A total of 123 up-regulated and 46 down-regulated proteins, and 12 up-regulated and 2 down-regulated metabolites were identified. The significant coexpression pathways were the HIF-1 (FDR = 1.86 × 10^−3^) and glycolysis/gluconeogenesis (FDR = 1.14 × 10^−10^). The results of validation analysis showed the proteins related to fluorine were mainly enriched in the cytoplasm and extrinsic component of the cytoplasmic side of the plasma membrane. The HIF-1 pathway (FDR = 1.01 × 10^−7^) was also identified. Therefore, the HIF-1 and glycolysis/gluconeogenesis pathways were significantly correlated with dental fluorosis.

## 1. Introduction

Fluorine is widely distributed in water, air, and soil in nature [1]. It is also one of the essential trace elements for human beings. Daily intake of small doses of fluoride can promote the development of teeth and bones [2,3]. However, long-term excessive intake of fluoride can lead to fluorosis, which results in damaged bone tissue [4], and its main clinical manifestations are skeletal fluorosis and dental fluorosis [5]. Hypoxia-inducible factor (HIF) is a heterodimeric protein that consists of oxygen-sensitive α-subunit (HIF-1α, HIF-2α, and HIF-3α) and a constitutively expressed β-subunit [6]. The functions of HIF-1α, HIF-2α, and HIF-3α are different. Of the three HIFα-homologs, HIF-1α is mainly responsible for regulating glycolysis. Gluconeogenesis is a process opposite to glycolysis, which occurs mainly in the liver. Of the 11 enzymes that catalyze the gluconeogenesis pathway, seven enzymes are identical to the glycolysis pathway, and the other four enzymes are unique to the gluconeogenesis pathway. The four enzymes are pyruvate carboxylase, phosphoenolpyruvate carboxykinase, fructose-1,6-phosphatase, and glucose-6-phosphatase.

Normally, the expression level of HIF-1α is very low. Under hypoxia conditions, the degradation amount of HIF-1α will be reduced and the level in the body is higher [7]. The activation of the HIF-1 signaling pathway will increase the expression of hexokinase and phosphofructokinase in the glycolysis/gluconeogenesis pathway, strengthen the glycolysis/gluconeogenesis pathway, and improve the yield of adenosine triphosphate to a certain extent [8]. In hypoxia, the glycolysis/gluconeogenesis pathway is mainly regulated by HIF-1α. The genes encoding glycolysis/gluconeogenesis enzymes, aldolase, beta-enolase (*Eno3*), and l-lactate dehydrogenase A chain (*Ldha*) contain HIF-1 binding sites in their enhancer regions and can be directly up-regulated by HIF. Glucose, as an important metabolic raw material for all living cells, can provide energy for cells in the body. When the glucose is transported into cells by glucose transporters (GLUTs), glucose-6-phosphate (G6P) is produced under the action of hexokinase. Then G6P generates pyruvate through an enzymatic reaction [9,10]. When the oxygen is sufficient in the cells, pyruvate is converted into acetyl coenzyme A under the action of pyruvate dehydrogenase, which enters the tricarboxylic acid cycle and is decomposed into carbon dioxide and water. However, when hypoxia occurs, pyruvate is converted into lactic under the action of lactate dehydrogenase, completing the process of glycolysis [6]. Both two different processes can provide energy to cells.

Enamel comes from the calcification of ameloblasts, which plays a key role in amelogenesis and is responsible for the synthesis, secretion, and hydrolysis of the organic matrix. Previous studies have confirmed dental fluorosis can occur in humans and animals exposed to fluoride during enamel maturation alone [11,12]. At this stage, the hydration is strong in the enamel matrix, and mineral and fluorine ions can freely enter the tooth germ. With the increased concentration of fluoride, the ameloblasts were more vulnerable and the degree of mineralization was lower [13,14]. Studies have shown that the process of enamel protein removal induced by fluoride [15,16], and mineral acquisition during enamel maturation [17] can make the crystal deposit too fast, and prevent the growth of crystals. As the nutrients and metabolites were exchanged between ameloblasts and their surrounding dense capillaries, long-term intake of fluoride at certain concentrations can enter ameloblasts through the peripheral blood, and metabolites from ameloblasts can also be measured in the peripheral blood [18].

However, few studies reported the differentially expressed proteins and metabolites in rats with dental fluorosis. This study was designed to investigate the mechanism of rats with dental fluorosis exposed to fluoride using integrated proteomics and metabolomics analysis.

## 2. Results

### 2.1. Identified Differentially Expressed Proteins and Metabolites

Compared with those in the control group, the rats in the dental fluorosis group occurred obvious pigmentation and chalk stripes on their teeth, indicating that the dental fluorosis model was successfully constructed (Figure 1b). Then 123 up-regulated and 46 down-regulated proteins (Appendix A), 12 up-regulated and 2 down-regulated metabolites (Appendix A) were identified. Figure 1c shows the differentially expressed proteins and Figure 1d shows the differentially expressed metabolites. Red dots represent the up-regulated differentially expressed proteins/metabolites, blue dots represent the down-regulated differentially expressed proteins/metabolites.

### 2.2. KEGG Pathway Enrichment Analysis

The KEGG pathway enrichment analysis determined the 33 coexpression pathways were significantly associated with dental fluorosis. The top 15 coexpression pathways included the biosynthesis of amino acids, glycolysis/gluconeogenesis, carbon metabolism, metabolic pathways, pentose phosphate pathway, phenylalanine metabolism, starch, and sucrose metabolism, starch and sucrose metabolism, alanine, aspartate and glutamate metabolism, tyrosine metabolism, and HIF-1 signaling pathway. HIF-1 and glycolysis/gluconeogenesis pathways were related to dental fluorosis (Figure 2a).

### 2.3. GO Function Enrichment Analysis

The GO enrichment analysis determined the top 10 biological processes involved in the small molecule metabolic process, organonitrogen compound metabolic process, response to stress, and single-organism developmental process. For the cell component, 53% of proteins existed in the extracellular region part. Its main molecular functions suggested that differentially expressed proteins and metabolites were related to anion binding, actin binding, carbon-nitrogen ligase activity, with glutamine as amido-N-donor, and enzyme inhibitor activity. (Figure 3).

### 2.4. KEGG Markup Language (KGML) Network Diagram

The KGML analysis showed that the HIF-1 signaling pathway contained five proteins named TFRC, RPS6, ENO3, LDHA, and ALDOA. Glycolysis/gluconeogenesis pathway covered 10 proteins named ALDH7A1, PGAM2, PGAM1, ALDOA, ALDOB, GPI, TPI1, PGM1, ENO3, and LDHA. These proteins/metabolites were mainly involved in the metabolism of sugars and energy (Figure 2b and Table 1).

### 2.5. Validation Analysis

In the Gene Cards, 136 fluorine-related genes were screened with a score greater than 1.5, and 145 KEGG pathways were enriched using the OE Biotech platform. The AGE-RAGE signaling pathway in diabetic complications and HIF-1 signaling pathway (FDR = 1.01 × 10^−7^) was significantly correlated with fluoride (Figure 4b). The GO analysis showed that the identified genes were mainly concentrated in the cytoplasm and extrinsic component of the cytoplasmic side of the plasma membrane (Figure 4a).

In the Human Protein Atlas Database, the PGAM1, ENO3, LDHA, GPI, ALDOA, PGM1, PGAM2, and TPI1 were highly expressed in muscle tissues and lowly expressed in the bone marrow. RPS6 and TFRC were highly expressed in the bone marrow. Finally, we explored the intracellular distribution of hub proteins in the U2-OS cell lines, the results showed that LDHA, GPI, PGAM2, ENO3, ALDOA, and RPS6 were mainly located in the cytosol. PGAM1 and TPI1 were mainly located in the nucleoplasm and TFRC was mainly located in the vesicles (Appendix A).

## 3. Discussion

Fluoride is widely present in the air, soil, and water. The fluoride is absorbed into the blood, and it quickly diffuses into various organs, and 60% to 80% is present in bone tissues. Due to the high affinity of fluorine to Ca^2+^, the calcium and fluoride in the blood can deposit in bone tissues in the form of CaF_2_, leading to a severe Ca^2+^ deficient environment in tooth germs, resulting in damaged ameloblasts and disordered enamel development. This study showed that the excessive intake of fluoride affects the expression of proteins and metabolites in the blood. Fluoride can alter the expression levels of nicotinamide, adenosine, 1-oleoyl-sn-glycero-3-phosphocholine, and 1-stearoyl-sn-glycerol 3-phosphocholine metabolites in the serum, leading to bone-related diseases [19]. The activation of the HIF-1 signal pathway and enhancement of glycolysis/gluconeogenesis are the signs of the activation of immune and inflammatory cells, and the metabolic pathways are changed in response to hypoxia [20]. A previous study showed that mitochondrial membrane potential could be affected by the activation of the HIF-1 signaling pathway and blood oxygen content [21]. In addition to the increased activity of enzymes in the glycolysis/gluconeogenesis pathway in response to hypoxia, the metabolites that entered the tricarboxylic acid cycle were inhibited by the HIF-1 pathway [22], which inhibited the oxidative phosphorylation (OXPHOS) [23,24].

In this study, the KEGG pathway enrichment analysis showed that the HIF-1 and glycolysis/gluconeogenesis pathways were related to dental fluorosis. The KGML network diagram showed that the proteins (LDHA, ENO3, GPI, and TPI1) enriched in the HIF-1 and glycolysis/gluconeogenesis pathways have a direct correlation with the formation of dental fluorosis. LDHA is responsible for converting pyruvate into lactate. It is mainly located in the cytoplasm, with a high pyruvate affinity [25]. LDHA can also be activated by HIF-1 [26]. Mikesh et al. found that LDHA was related to the respiratory intensity of mitochondria [27]. Enolase, composed of three subunits (α, β, and γ), catalyzes the glycolysis/gluconeogenesis step interconverting 2-phosphoglycerate and phosphoenolpyruvate. The β-subunit is encoded by the ENO3 gene and mainly exists in the cytoplasm, nucleus, and cell surface [28]. GPI is a housekeeping cytosolic enzyme encoded by the *Gpi* gene, catalyzing the interconversion between G6P and fructose-6-phosphate, the second reaction step of glycolysis [29].TPI1 is a glycolysis/gluconeogenesis enzyme, which is responsible for catalyzing the isomerization of dihydroxyacetone phosphate (DHAP) and glyceraldehyde-3-phosphate (G3P), the G3P is an intermediate of glycolysis/gluconeogenesis [30]. TPI1 is located in the cytoplasm and comes after fructose-bisphosphate aldolase. Aldolase can generate DHAP and G3P [31]. The up-regulation of TPI1 accelerates the generation of G3P from DHAP, promoting the metabolism of DHAP [30].

The up-regulation of ENO3, LDHA, TPI1, and GPI implies that the HIF-1 pathway and glycolysis/gluconeogenesis pathway are strengthened in ameloblasts, which affects the energy harvesting of ameloblasts and leads to the formation of dental fluorosis. In response to hypoxia, the glycolysis/gluconeogenesis pathway is improved by the HIF-1 signaling pathway. Meantime, the OXPHOS mediated by the HIF-1 pathway was suppressed to increase glycolytic flux. The mitochondria in ameloblasts were damaged by fluoride, aerobic respiration was reduced, and the glycolysis/gluconeogenesis pathway was enhanced [32].

Dental fluorosis is induced by high concentrations of fluoride during the formation of enamel, which is characterized by discolored, mottled, and porous. Enamel is a type of hydroxyapatite secreted by ameloblasts. Studies have shown that fluoride can affect the formation and degradation of enamel matrix proteins [33]. Disturbance of ameloblast energy acquisition can affect the formation of enamel. The enhancement of anaerobic respiration often means damage to aerobic respiration, reducing energy acquisition. The enhancement of the HIF-1 and glycolysis/gluconeogenesis pathways means that the anaerobic respiration of ameloblasts is enhanced, and the enamel secretion is affected, which leads to the formation of dental fluorosis.

The GLUT1 mainly exists on the surface of the cell membrane and is responsible for glucose transport. When the expression of HIF-1α increases, the transcription of GLUT1-related proteins is regulated and the expression of GLUT1 is enhanced. When blood glucose concentrations rise, insulin secretion will be increased. Then stimulated GLUT4 is transferred to the cell surface, and transports glucose along with GLUT1 [34]. Inhibitors of apoptosis proteins (IAPs) are a class of apoptosis regulatory proteins that protect cells from apoptosis by regulating caspase activation and NF-κB signaling transduction factors. Both IAPs and glycolysis pathways are highly expressed in cancer cells compared with those in normal cells [35,36].

Long-term intake of fluoride can also change the gut microbiome in the body. The gut microbiome can maintain the health and stability of metabolites in the body by influencing the production of short-chain fatty acids, the synthesis of essential vitamins, and the differentiation of T lymphocytes [37,38]. The intake of foreign stimulants also affects the body’s microbiome, which in turn affects the expression of metabolites.

According to our findings, we can selectively develop nutritional supplements that can enhance aerobic respiratory strength of antioxidant stress to protect or save the health of fluorosis patients. This study has some limitations. For example, only six dental fluorosis rats were included in this study, and the number of samples was small. The direct distribution and expression of these differential proteins in ameloblasts have not been conducted, and the conclusion lacks certain persuasion. However, in subsequent studies, we will consider increasing the number of samples and providing direct evidence on ameloblasts.

## 4. Materials and Methods

### 4.1. Sample Collection and Preparation

All procedures were performed in accordance with national and institutional animal care guidelines and were approved by the Ethics Committee of Zhengzhou University (ZZUIRB 2019-012). The 30 specific pathogen-free (SPF) Sprague Dawley (SD) rats were purchased from Henan Experimental Animal Center and raised in the SPF environment of the animal laboratory of Zhengzhou University. They were randomly divided into two groups according to gender and body weight (ten female and five male rats). After adaptive feeding for 1 week, the rats in the control group were treated with pure water, and those in the dental fluorosis group were treated with 100 mg/L sodium fluoride. All animals had free access to food and water. The male and female rats in each group were fed in the cages at a 2:1 ratio (two female and one male rats in one cage). The male rats in the same group were replaced every other day, and the pregnant rats were observed every day. After the delivery of female rats, the offspring rats and female rats were placed in the same cage until 21 days postpartum. After weaning, eight offspring rats in the control group and six offspring rats in the dental fluorosis group were randomly selected to maintain the same interventions as their parents. The offspring rats were anesthetized in the third month. All rats were housed under standard laboratory conditions (18–22 °C; 12 h light: 12 h dark cycle) and euthanized using the cervical vertebra dislocation method. The abdominal aortic blood was extracted from the abdominal aorta and placed in anticoagulant and nonanticoagulant tubes, respectively. The samples were stood for 30 min, and centrifuged at 3000× *g* r/min for 10 min, the upper plasma was separated into a centrifuge tube and stored in a −80 °C refrigerator.

### 4.2. Protein and Metabolite Extraction

#### 4.2.1. Proteins Extraction

The 40 μL blood samples were collected and diluted with ×10 binding buffer; then the diluted blood samples were added to OASIS HLB and allowed to flow through the column by gravity. The OASIS HLB was washed with 600 μL binding buffer, and the extracting solution was collected and albumin/IgG was removed. Moreover, the sample was freeze-dried in a vacuum for use. The lyophilized samples were reconstituted by adding 300 μL SDS to lyse and centrifuged at 12,000× *g* for 10 min at room temperature, and the supernatant was collected. The concentration of extracted proteins was determined using the bicinchoninic acid method. The SDS-polyacrylamide gel electrophoresis was used to measure the quality of protein. Then, 50 μg protein was taken from each sample, and dithiothreitol (5 mM) was added to the protein solution and the mixture was incubated at 55 °C for 30 min. The iodoacetamide (10 mM) was added, and the mixture was kept placed in the dark for 15 min at room temperature. Then acetone (6 times the volume of solution) was added to the above solution to allow the proteins to precipitate overnight. The precipitates were collected at 8000× *g* at 4 °C for 10 min. Then 100 μL TEAB (200 mM) was added to reconstitute the precipitate, 1 mg/mL of Trypsin-TPCK was added at 1/50 sample mass, and the solution was digested at 37 °C overnight. Next 50 μL TEAB buffer (100 mM) was added to the sample, mixed by vortexing, and the reaction was labeled in a 1.5 mL EP tube. Then, 41 μL TMT reagent was added to the Ep tube, and the mixture was left at room temperature for 1 h. The reaction was terminated by adding 8 μL hydroxylamine (5%) for 15 min and stored at −80 °C after lyophilization. Then peptide labeled proteins were analyzed by liquid chromatography with tandem mass spectrometry (LC-MS/MS).

#### 4.2.2. Metabolites Extraction

Blood samples were collected into a 1.5 mL EP tube, with the l-2-chlorophenyl alanine (0.3 mg/mL) dissolved in methanol as the internal standard, and rotated for 10 s. An ice-cold mixture of methanol and acetonitrile (2/1, *v*/*v*) was then added; the mixture was rotated for 1 min, ultrasonic extracted in an ice water bath for 10 min, and centrifuged for 10 min (13,000× *g* rpm) at 4 °C. The supernatant was dried in a freeze concentration centrifugal dryer. A pooled sample was prepared by mixing an aliquot of all samples. Then an equal amount of supernatant was transferred to a glass bottle and dried under a vacuum at room temperature. Then, the pyridine solution of methoxyamine hydrochloride was added. The mixture was rotated violently for 2 min and incubated for 90 min at 37 °C. Then, BSTFA (containing 1% TMCS) and n-hexane were added to the mixture, which was then rotated violently for 2 min and derivatized at 70 °C for 60 min. The samples were kept for 30 min at room temperature and analyzed by gas chromatography-mass spectrometry (GC-MS).

### 4.3. LC-MS/MS and GC-MS Analyses

#### 4.3.1. Differentially Expressed Proteins

The peptide-labeled proteins were analyzed using an Agilent 1100 HPLC. In the process of reverse phase chromatographic separation, mobile phase A: ACN-H_2_O (2:98, *v*/*v*) and mobile phase B: ACN-H_2_O (90:10, *v*/*v*). The pH of mobile phases A and B were adjusted to 10 using ammonia water. The flow rate was set at 300 μL/min. The gradient elution conditions were as follows: 0–8 min, 98% A; 8–8.01 min, 98–95% A; 8.01–48 min, 95–75% A; 48–60 min, 75–60% A; 60–60.01 min, 60–10% A; 60.01–70 min, 10% A; 70–70.01 min, 10–98% A; 70.01–75 min, 98% A. The eluates were collected in centrifuge tubes no. 1–15 and the samples were cryopreserved for MS. The chromatographic conditions were as follows: the sample was loaded onto the precolumn at a flow rate of 350 μL/min and then separated by an analytical column (RP-C18, New Objective, MA, USA), mobile phase A: H_2_O-FA (99.9:0.1, *v*/*v*) and mobile phase B: ACN-FA (99.9:0.1, *v*/*v*). The gradient elution conditions were as follows: 0–1 min, 2–6% B; 1–52 min, 6–35% B; 52–54 min, 35–90% B; 54–60 min, 90% B.

#### 4.3.2. Differentially Expressed Metabolites

The derived samples were analyzed using the Agilent 7890B GC system coupled with an Agilent 5977B MSD system (Agilent Technologies Inc., Santa Clara, CA, USA). An HP-5MS fused-silica capillary column (Agilent J&W Science, Folsom, CA, USA) was used to separate the derivatives. The carrier gas was helium (>99.999%) and the flow rate was set at 1 mL/min. The injector temperature was maintained at 260 °C. The initial oven temperature was 60 °C, which was maintained for 0.5 min, increased to 125 °C at a rate of 8 °C/min, increased to 210 °C at a rate of 5 °C/min, increased to 270 °C at a rate of 10 °C/min, increased to 305 °C at a rate of 20 °C/min and finally maintained at 305 °C for 5 min. The temperatures of the MS quadrupole and the ion source (electron incidence) were set to 150 and 230 °C, respectively. The collision energy was 70 eV. The pooled samples were injected at regular intervals throughout the analysis, to provide a dataset that can be used to evaluate repeatability.

### 4.4. Identified Differentially Expressed Proteins and Metabolites

The differentially expressed proteins were measured using LC-MS/MS; the library was searched, and the proteins were quantitatively and qualitatively analyzed. Then, raw data were retrieved from the database and credible proteins were screened according to the following criteria: score sequence HT > 0 and unique peptide ≥ 1. Blank values were removed. The differentially expressed proteins were identified based on the following criteria for difference screening: Fold change >1.2 or <0.83 and *p*-value < 0.05.

The obtained GC-MS raw data were converted into .abf format; then, the data were imported into MS-DIAL software. Metabolite characterization was based on the LUG database. A data matrix was derived. The three-dimensional matrix included sample information, the name of the peak of each substance, retention time, retention index, mass-to-charge ratio, and signal intensity. In each sample, all peak signal intensities were segmented and normalized according to the internal standards with RSD ≥ 0.3 after screening. The obtained metabolites were identified using two-tailed Student’s *t*-test. The screening criteria for differential metabolites were as follows: variable importance of projection (VIP) value > 1.0 and *p*-value < 0.05.

### 4.5. Integrated Proteomics and Metabolomics Analyses

The integrated proteomics and metabolomics analysis was performed using OmicsBean (http://www.omicsbean.cn/; accessed on 28 January 2021). OmicsBean is a practical and efficient omics data integration analysis tool, including quality control analysis, difference analysis, Venn group screening, GO analysis (biological process, cell component, and molecular function), KEGG pathway analysis, and molecular mechanism model construction. The results are displayed in the dynamic charts, which can be customized for editing and downloading. Meanwhile, multiple omics combined analyses can also be performed. This study was designed to identify the coexpressed KEGG pathways and GO of differentially expressed proteins and metabolites using multiple omics analysis modules, and FDR < 0.05 was considered significant.

### 4.6. Validation of Database

The fluoride-related genes were collected from the Gene Cards (https://www.genecards.org/; accessed on 18 April 2022). The relevance scores ≥ 1.5 were set as the screening criteria. GO and KEGG pathways were analyzed using the OE Biotech platform (https://cloud.oebiotech.cn/task/; accessed on 18 April 2022). The Human Protein Atlas (HPA) Database (https://www.proteinatlas.org/; accessed on 27 May 2022) was used to determine the subcellular distribution of key proteins and their expression in bone marrow, blood, and muscle tissues [39].

## 5. Conclusions

Fluoride can affect the energy acquisition of ameloblasts by upregulating the activities of enzymes that are enriched in glycolysis/gluconeogenesis and HIF-1 signaling pathways, resulting in the development of dental fluorosis. This may provide new insights for treating dental fluorosis.

## Figures and Tables

**Figure 1 ijms-23-08266-f001:**
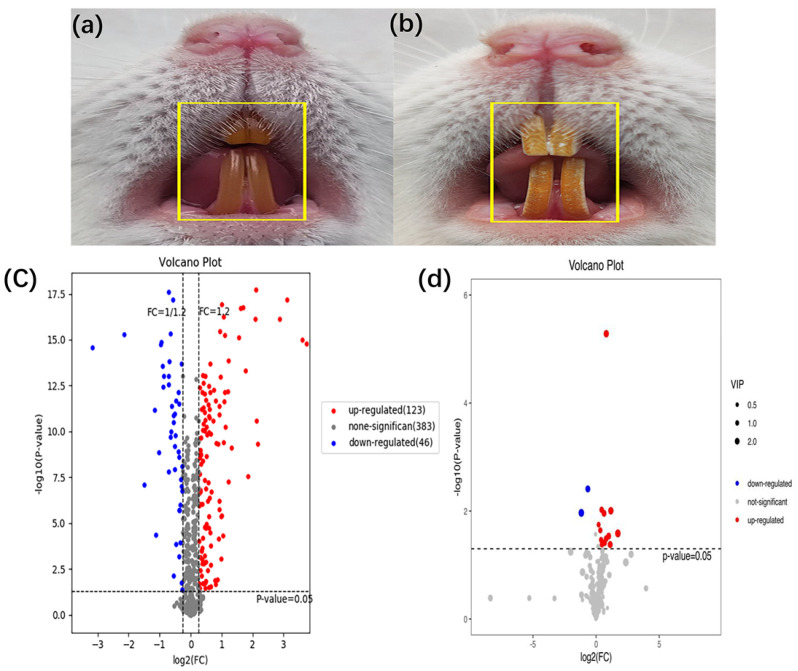
Establishment of animal model for dental fluorosis, (**a**) shows normal dental of mice (*n* = 8) and (**b**) shows dental fluorosis of mice (*n* = 6). The differentially expressed proteins and metabolites extracted from peripheral blood using the liquid chromatography tandem mass spectrometry and gas chromatography mass spectrometry, the fold change >1.2 or <0.83 and *p* < 0.05 were considered significant. A total of 169 differentially expressed proteins and 14 differentially expressed metabolites were obtained. The volcano of proteomics (**c**) and metabolomics (**d**) in the dental fluorosis group and control group.

**Figure 2 ijms-23-08266-f002:**
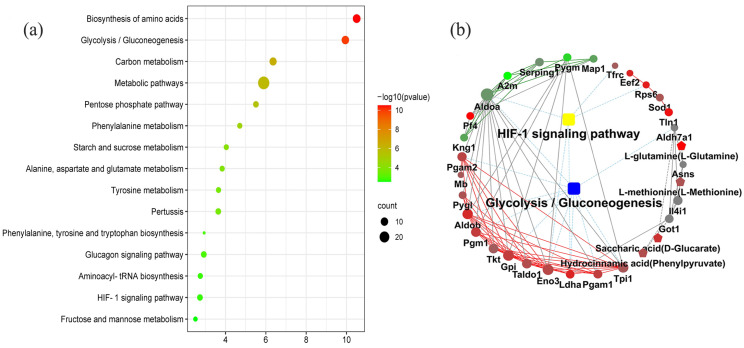
The top 15 coexpression KEGG pathways integrated proteomics and metabolomics using the OmicsBean, FDR < 0.05 (**a**). Proteins and metabolites interaction network diagram obtained from the OmicsBean platform show the differentially expressed proteins and metabolites related to the HIF-1 signaling pathway and glycolysis/gluconeogenesis pathway in the dental fluorosis group and control group (**b**). The samples included 6 dental fluorosis rats and 8 normal dental rats.

**Figure 3 ijms-23-08266-f003:**
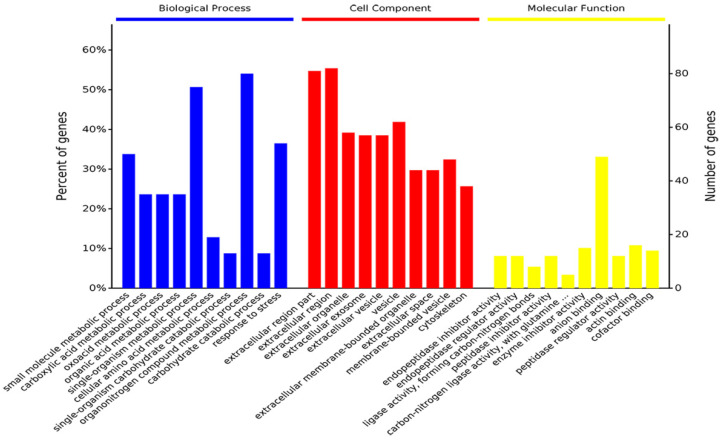
Gene ontology (biological process, cellular component, and molecular function) enrichment analysis was conducted between proteomics and metabolomics. The samples included 6 dental fluorosis rats and 8 normal dental rats.

**Figure 4 ijms-23-08266-f004:**
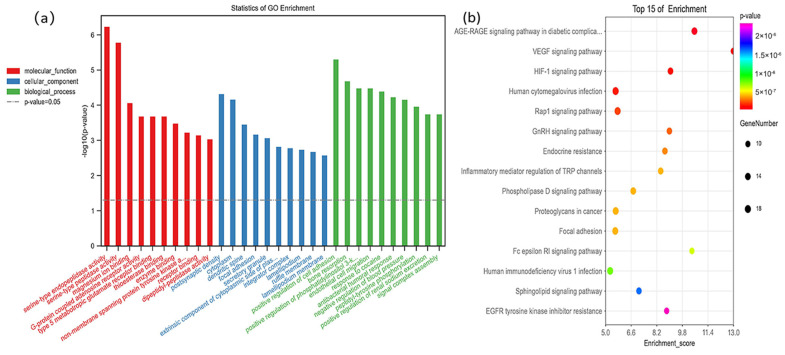
The top 10 gene ontology (**a**) and top 15 KEGG pathways (**b**) enrichment analysis of fluoride-related genes using oebiotech, and the fluoride-related genes are obtained from Gene Cards (https://www.genecards.org/; accessed on 18 April 2022) with a score greater than 1.5.

**Table 1 ijms-23-08266-t001:** The differential expressed proteins enriched in the glycolysis/gluconeogenesis and HIF-1 signaling pathway.

Gene Names	Description	Accession	*p*-Value	Fold Change
*Ldha*	l-lactate dehydrogenase A chain	P04642	2.14 × 10^−2^	1.74
*Aldob*	Fructose-bisphosphate aldolase	Q66HT1	2.63 × 10^−11^	1.67
*Pgm1*	Phosphoglucomutase-1	P38652	1.12 × 10^−10^	1.50
*Gpi*	Glucose-6-phosphate isomerase	Q6P6V0	5.78 × 10^−8^	1.50
*Pgam1*	Phosphoglycerate mutase 1	P25113	6.13 × 10^−7^	1.48
*Pgam2*	Phosphoglycerate mutase 2	P16290	2.20 × 10^−9^	1.43
*Eno3*	Beta-enolase	P15429	1.35 × 10^−7^	1.40
*Tpi1*	Triosephosphate isomerase	P48500	2.95 × 10^−4^	1.29
*Aldoa*	Fructose-bisphosphate aldolase A	P05065	3.32 × 10^−11^	0.69
*Tfrc*	transferrin receptor	Q99376	1.03 × 10^−9^	1.25
*Rps6*	ribosomal protein s6	P62755	1.18 × 10^−2^	1.82

## Data Availability

The data that support the findings of this study can be found in Appendix A.

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
