# Peer review of "Role of Glycolysis/Gluconeogenesis and HIF-1 Signaling Pathways in Rats with Dental Fluorosis Integrated Proteomics and Metabolomics Analysis"

_ijms, 2022, doi:10.3390/ijms23158266_

Round 1

Reviewer 1 Report

The manuscript of Be et al describes metabolic consequences in blood after excess fluoride in diet. Metabolomics and Proteomics results were correlated to build a big picture view on changes. Authors describe some of the main hits and discuss their potential roles in dental fluorosis. 

I have a few comments on this manuscript that should be addressed before I can recommend this manuscript for publication. Comments are under the following main categories:

1.     Description of results

2.     Validation of targets

3.     Discussion on relevance

4.     Figures

Here more in detail to each point.

1.     Throughout the text and figure legends authors should make sure methods are clearly described. For example, in abstract, authors mention metabolic measurements were performed but do not specify in which tissue/biofluid. These details are needed for the reader to follow the text and logic of the experiments. Further, figure legends are missing key details on methods used. Especially for Figure 4, authors must indicate that this is not primary data and cite relevant literature.

Lines 13-14: please specify biofluid/tissue measured

Lines 76-78: please specify biofluid/tissue measured; proteomics and metabolomics methods details, statistics thresholds used; number or proteins and number of metabolites detected/measured

Lines 88-90: please specify details on methods and statistics. Please indicate significant p-value threshold for (a). please include details on method for (c).

Lines 123-124: please specify details on how data was generated and analyzed.

Line 126: Please specify details on where the data was take from, cite relevant literature and include details on what each color on the figure represents (b) and in which tissue or cell-line the results were generated (a)

Furthermore, authors should prepare supplementary tables with proteomics and metabolomics data results. Each detected protein/metabolite should be indicated and quantitative data reported in a table format. Could the authors also include what proteomics/metabolomics libraries were queried for annotation?

2.      Authors report on global metabolomics and proteomics changes. However, it will be informative if authors could validate proteomics results using Western blot. Authors could pick top changed candidate proteins, especially those discussed in further detail in the text, and probe with specific antibodies to validate their targets. Do the changes occur only in blood or in other tissues? What is the relevant tissue for fluorosis?

3.     I believe the discussion could be improved. Key points not directly addressed in the text should be discussed: is the measurement of blood proteins and metabolites relevant to fluorosis and could changes in blood indicate other, for example liver-toxicity, effects of fluoride on the whole body, rather than changes associated with dental fluorosis? Authors could also discuss how they see the role of HIF-1 and glycolysis specifically in dental fluorosis – effect on crystal deposition, effect on ameloblasts and time of action, effects on adult or embryonic stages etc.

4.     I find the figures particularly hard to read. Authors could make sure that every panel is legible. Panels should not be disproportionally stretched and text should be big enough to be read by eye after printing. Text on figures could also be simplified to help presentation and interpretation of results. Also, figures should be of sufficient resolution for the text to be legible. Finally, it should be made clear that figure 4 is not based on primary data and relevant articles must be cited (in figure legend in addition to methods section).

Finally, I strongly encourage the authors to proofread the manuscript for grammatical and logical irregularities.

Reviewer 2 Report

In the present study, Yue Ba and coworkers experimentally investigated the mechanism of rats with dental fluorosis exposed to fluoride using integrated proteomics and metabolomics analysis. The authors concluded that the HIF-α and glycolysis/gluconeogenesis signaling pathway were significantly correlated with dental fluorosis.

Overall, I think the paper is timely, interesting and the data are of potential clinical relevance.

However, in my opinion, the present paper could be further improved, and I make some suggestions.

1) GLUT-1 and GLUT-4 seem to be involved in the modulation of explored signaling pathway. This feature should be considered in the paragraph of discussion.

2) Inhibitors of apoptosis (IAPs) could have an emerging role in  glycolysis/gluconeogenesis mechanism. Please make an appropriate comment about it.

3) Current evidence has shown the involvement of the gut microbiome on molecular signaling pathway of gut/metabolic disorders. Please make a comment in the discussion section of revised manuscript on a hot topic of current research.

4) In light of the results obtained in the present study, please to discuss on the possible application of nutraceutics and/or antioxidants/antinflammatory compounds, that, could also provide a potential therapeutic strategy in the treatment of dental fluorosis.

Reviewer 3 Report

Introduction:

1.     Lines 32-34: The role of HIF-3α is different from that of HIF-1α and HIF-2α.

2.     Lines 36-41: The role of HIF in regulation of glycolysis and gluconeogenesis should be more clearly described. Most cells are not even capable of gluconeogenesis. Moreover, not all enzymes of glycolysis and gluconeogenesis are shared by both pathways (e.g. PFK is not shared).

3.     Lines 46-48: The role of pyruvate dehydrogenase, TCA, and oxidative phosphorylation should be described more precisely. As written, it is not entirely correct.

4.     Line 49: Strictly speaking. LDH converts pyruvate into lactate and not lactic acid.

Methods:

5.     Methods lack important information, such as description of cell culturing conditions (U2-OS cell lines). Techniques used for analysis of cell cultures (Figure 4) should be described. Statistical methods should also be described. In addition, procedures on animals should be described in more detail, including euthanasia.

Results:

6.     Lines 97-99: It is not clear what the following sentence means: “The HIF-1 single pathway and glycolysis/gluconeogenesis pathway occur in the extracellular cytoplasmic matrix and involves the metabolism of organic acids and organic substances (Figure 2b).” How can it occur in the extracellular matrix?

7.     Results should demonstrate an upregulation of HIF-1 in tissue samples. Also, the results should describe more clearly which samples were used to obtain results. Based on the description in the methods it appears only blood samples were used, but this is not clear from the description of results. How does this relate to the pathology of the teeth?

8.     Figure 4: how do these results contribute to understanding of pathogenesis of fluorosis? How were the experiment and analyses carried out?

9.     How does 100 mg/L NaF relate to the typical daily dose of fluoride that leads to fluorosis in humans? NaF is a well-known metabolic poison, which may be the explanation for the observed effects. What is the evidence that the observed changes are important for fluorosis?

10.  Figure legends should clearly indicate how many samples were used for analyses.

11.  Lines 116-118: “In gene expression, Pgam1, Eno3, Ldha, Gpi, Aldoa, Pgm1, Pgam2, and Tpi1 were highly in muscle tissues and lowly expressed in bone marrow. Rps6 and Tfrc were highly expressed in bone marrow.” It is unclear from where these data were obtained.

Discussion:

12.  Discussion should be more developed. It should refer to other studies and put the current study in the context of existing knowledge about pathogenesis of fluorosis. It should also describe what does the current study add to existing knowledge, limitations of the study etc.

Conclusions:

13.  “KGML network diagram showed that the proteins (Ldha, Eno3, Gpi, and Tpi1) played an important role in the formation of dental fluorosis.” This is just a correlation not a proof of causation.

14.  “Fluoride can affect the energy acquisition of ameloblasts by up regulating the activities of enzymes that enriched in glycolysis/gluconeogenesis and HIF-1, resulting in the formation of dental fluorosis” Based on the description in the methods it appears that only blood was analysed. Results therefore do not directly indicate what happened in ameloblasts. Is there any evidence that response of ameloblast might be similar?

15.  “This may provide a new idea for the prevention and treatment of dental fluorosis.” Prevention of fluorosis likely involves lower fluoride ingestion rather than modulation of HIF pathways. So do these results really indicate how to prevent fluorosis (since prevention is lower intake of fluoride)? Also, how do these results indicate new therapeutic possibilities?

Round 2

Reviewer 1 Report

thank you. no further comments

Author Response

Many thanks for your helpful comments.

Reviewer 2 Report

The authors have satisfactorily responded to all my questions and made the necessary changes to the manuscript.

Author Response

Many thanks for your helpful comments.

Reviewer 3 Report

Improvements and replies by the authors are appreciated. However, there are still some outstanding issues:

1.     The relevance of studying blood cells in the context of fluorosis should be discussed in more detail. For instance, is it possible to extrapolate data from blood cells to ameloblasts? Is there any published evidence that analysis of blood cells provides an indication of alterations in ameloblasts? Or do perhaps alterations in the blood reflect some other pathological processes that occur in fluorosis? These issues should be more clearly presented in the introduction and in the discussion.

2.     “Under hypoxia conditions of hypoxia, the HIF-1 signaling pathway will be is activated and the expression of HIF-1α will be increased [7].” This sentence could be misunderstood. The main effect in hypoxia is stabilization of HIF-1α protein due to reduced proteasomal degradation. Gene expression may change, but the main effect is on a posttranslational level.

3.     Authors should check whether abbreviations for human genes/proteins are written in a correct way. For instance, human Pgam1 should probably be PGAM1 (protein) or PGAM1 (gene)? Similarly, animal genes should probably be written in italics Pgam1.

4.    It is unclear how Fig. 5 contributes to understanding of fluorosis. RNA data and images of U-2 OS cells in Fig. 5 are all from an on-line database, it would be better not to include them in the results.  In any case, the cellular distribution of classical enzymes, such as LDH, under basal conditions does not need to be provided as a result, because these are all known facts. 

5.     “When blood glucose concentrations rise, GLUT4 is transferred to the cell surface to transport glucose along with GLUT1 [34].” Translocation of GLUT4 is stimulated by insulin and not high glucose.

6.     “the HIF-1 signaling pathway contained five proteins/metabolites named Tfrc, Rps6, Eno3, Ldha, and Aldoa.” None of these is a metabolite (i.e. just proteins are mentioned).

7.     “…glycolysis/gluconeogenesis signaling pathways…” It is not clear what is meant by glycolysis/gluconeogenesis signalling.

8.     Interconversion of pyruvate and lactic acid is still mentioned (lines 175/176).

9.     U-2 OS cells are osteosarcoma cells – why are they relevant for pathological process that affects ameloblasts?

Round 3

Reviewer 3 Report

The authors have replied to my questions and suggestions. I have just two final suggestions:

(1)   In section 2.4 (of the Results) proteins/metabolites are mentioned (“Glycolysis/gluconeogenesis pathway covered 10 proteins/metabolites named ALDH7A1, PGAM2, PGAM1, ALDOA, ALDOB, 124 GPI, TPI1, PGM1, ENO3, and LDHA. These proteins/metabolites were mainly involved in the metabolism of sugars and energy (Figure 2b and Table 1)”, but again no metabolites are listed. Indeed, all of them are proteins, so the text should be corrected accordingly.

(2)   Regarding the glycolysis/gluconeogenesis signalling pathway I suggest to the authors to use the expression metabolic pathway instead of signalling pathway. Signalling suggests signalling cascades, involving receptors, protein kinases, G-proteins,… Since the authors actually refer to metabolism it would be more correct to write glycolysis/gluconeogenesis pathway (or glycolysis/gluconeogenesis metabolic pathway, but not “glycolysis/gluconeogenesis signalling pathway”.
